# A New Definition of the Stationary Phase Volume in Mixed-Mode Chromatographic Columns in Hydrophilic Liquid Chromatography

**DOI:** 10.3390/molecules26164819

**Published:** 2021-08-09

**Authors:** Pavel Jandera, Tomáš Hájek

**Affiliations:** Department of Analytical Chemistry, University of Pardubice, Studentská 573, 53210 Pardubice, Czech Republic; Pavel.Jandera@upce.cz

**Keywords:** hydrophilic interaction, liquid chromatography, stationary and mobile phase, polar columns

## Abstract

Polar columns used in the HILIC (Hydrophilic Interaction Liquid Chromatography) systems take up water from the mixed aqueous–organic mobile phases in excess of the water concentration in the bulk mobile phase. The adsorbed water forms a diffuse layer, which becomes a part of the HILIC stationary phase and plays dominant role in the retention of polar compounds. It is difficult to fix the exact boundary between the diffuse stationary and the bulk mobile phase, hence determining the column hold-up volume is subject to errors. Adopting a convention that presumes that the volume of the adsorbed water can be understood as the column stationary phase volume enables unambiguous determination of the volumes of the stationary and of the mobile phases in the column, which is necessary for obtaining thermodynamically correct chromatographic data in HILIC systems. The volume of the aqueous stationary phase, Vex, can be determined experimentally by frontal analysis combined with Karl Fischer titration method, yielding isotherms of water adsorbed on polar columns, which allow direct prediction of the effects of the composition of aqueous–organic mobile phase on the retention in HILIC systems, and more accurate determination of phase volumes in columns and consistent retention data for any mobile phase composition. The n phase volume ratios of 18 columns calculated according to the new phase convention strongly depend on the type of the polar column. Zwitterionic and TSK gel amide and amine columns show especially strong water adsorption.

## 1. Introduction

High Performance Liquid chromatography (HPLC) has become one of the most powerful tools for the separation and determination of samples containing non-polar, moderately or strongly polar and ionic compounds, either simple species or complex high-molecular synthetic polymers or biopolymers. Non-ionic compounds are usually separated on the basis of differences in polarities, most often in reversed-phase (RP) systems with a non-polar stationary phase and a mixed aqueous–organic mobile phase. For the separation of moderately polar compounds, the conventional HPLC employs a polar adsorbent such as silica gel and a less polar mixed organic solvent mobile phase in organic normal-phase (NP) systems. However, strongly polar and partially ionized compounds are often weakly retained and poorly resolved in the RP systems and much too strongly (often irreversibly) retained in the NP systems. The issue was significantly alleviated by introduction of Hydrophilic Interaction Liquid Chromatography (HILIC) employing polar columns and mobile phases with high concentrations of polar organic solvents in water, i.e., aqueous normal-phase systems. Various polar columns may be suited for this purpose, including fully porous and core-shell silica gel, hybrid organic-silica, silica hydride, chemically bonded polar stationary phases with bonded cholesteryl, phenyl, nitrile, pentafluorophenyl propyl, diol, zwitterionic sulfobetaine, phosphorylcholine and other ligands. HILIC has been found especially useful in (not only) pharmaceutical, biomedical and clinical analysis.

Polar adsorbent surface has strong affinity towards water, which it takes up from aqueous–organic mobile phases. There is general agreement that the water adsorbed in concentrations exceeding the volume fraction in the bulk mobile phase is the main origin of the retention of polar compounds in HILIC [1]. The computer molecular dynamics simulation studies indicate that the relative proportion of the amount of water contained in the pores of silica-based phases to the water concentration in the bulk mobile phase increases at low total water concentrations in the column [2] This may be the reason why the LC separations are often irreproducible or fail in the mobile phases containing less than 2% water in acetonitrile. The water molecules close to the silica surface strongly adhere to the silanol groups by the hydrogen bonds. Three types of water molecules co-exist inside the 6–10 nm pores: free water molecules, “freezable” bound water, and bound water that does not freeze at the regular water freezing temperature [3].

Because water is miscible at any proportion with acetonitrile, acetone, methanol or other polar organic solvents used in the HILIC mode, the adsorbed water forms a diffuse layer lacking fixed boundaries. The concentration of water progressively decreases from the polar solid surface towards the bulk organic-rich mobile phase outside (possibly, even partly inside) the pores of the stationary phase [4]. The retention mechanism in HILIC depends on the polarity of the column material. The adsorbed water layer plays principal role in the retention mechanism.

In HPLC, the retention time is controlled by the ratio of the time spent by a solute in the stationary phase, *t_s_*, to the time spent in the mobile phase, *t_m_*, i.e., the retention factor, *k*, which is equal to the ratio of the masses of the solute in the stationary, *m_s_*, and in the mobile, *m_m_*, phases, in the column. *k* is directly proportional to the constant, *K_D_*, characterizing the distribution of the solute between the stationary and the mobile phase:(1)$$k=\frac{t_{s}}{t_{m}}=\frac{t_{R}-t_{m}}{t_{m}}=\frac{V_{R}-V_{m}}{V_{m}}=\frac{m_{s}}{m_{m}}=\frac{c_{s}}{c_{m}}\cdot \frac{V_{s}}{V_{m}}=K_{D}\frac{V_{s}}{V_{m}}=K_{D}\Phi$$

The retention factor, *k*, depends on the nature of the solute, of the stationary and the mobile phases and on temperature, but is independent of the flow rate of the mobile phase, the dimensions of the column (provided equal density of packing, i.e., a constant phase ratio along the column). Hence, *k* is a fundamental parameter in the method development and optimization of HPLC separations and for measuring thermodynamic quantities of the retention process. Unfortunately, the retention factors calculated from the experimental retention data using Equation (1) do not provide reliable information on the presumed mechanism of retention, as several different mechanisms may contribute to the actual *k*.

Ideally, the equilibrium distribution of the sample compounds between the stationary and the mobile phase instantaneously re-establishes at any time in any part of the chromatographic bed. The changes in the partial molar Gibbs free energy, Δ*G*, of the solute transfer from the mobile to the stationary phase control the thermodynamics of the chromatographic for strongly diluted samples:Δ*G* = −*RT* ln *K_D_* = −*RT* ln (*c_s_* /*c_m_*)(2)

*R* is the gas constant, *T* is the temperature (in Kelvins) and *K_D_* is the distribution (partition) constant, which gives the equilibrium ratio of the concentrations of the solute in the stationary, *c_s_*, and in the mobile, *c_m_*, phases, respectively [5]. 

The proportionality constant Φ in Equation (1) is the phase ratio, i.e., the ratio of the volumes of the stationary, *V_s_*, and of the mobile, *V_m_*, phases in the column. From the Equation (2), it follows that:(3)$$t_{R}=t_{m}\left(1+k\right)=\frac{L}{u}\left(1+k\right)$$
(4)$$V_{R}=V_{m}\left(1+k\right)=t_{m}\cdot F\left(1+k\right)$$

*t_m_* and *V_m_* are also known as the column hold-up time and hold-up volume, respectively. The terms *t_0_*, and *V*_0_ are sometimes used instead of *t_m_* and *V_m_*. The ratio of the column length, *L*, to the column hold/up time, *t_m_,* controls the linear velocity of the mobile phase along the column, *u*. *F* is the volume flow rate of the mobile phase, a simple conversion factor between the retention times and retention volumes [6].

For correct determination of the retention factors, we must know the volumes of the stationary and mobile phases in the column, which determine the column phase ratio, Φ = *V_s_*/*V_m_* in Equation (1).

Simple estimation of the column hold-up time, *t_0_*, as the time of the appearance of the first disturbance peak on the detector baseline may be subject to serious errors. A more sophisticated estimation involves setting the void volume equal to the elution volume of an “inert” compound, which neither taken up nor excluded from the solid phase (after the correction for extra-column volumes). The selection of a suitable marker compound is generally pragmatic and may not always yield the correct *V_m_* values. The marker selection is especially challenging in HILIC systems. Small polar molecules such as uracil or thiourea are often used as the column markers in reversed-phase chromatography, non-polar hydrocarbons such as benzene or toluene in the HILIC systems [4]. However, the elution times of benzene and toluene on silica gel columns may slightly increase at 30% or more water in the mobile phase. This means that the water amount adsorbed close to the polar adsorbent surface depends on the water concentration in the bulk mobile phase [7]. Even the retention time of a component of the mobile phase or of a monovalent salt may not provide accurate *V_m_* values [8]. Integrating the plots of the retention times of the perturbation peak from 0 to 100 % of the organic modifier [9] is time consuming and the peak is not always clearly apparent, or may be due to the impurities excluded before the real column hold-up time [10]. Static methods of the determination of the column hold-up volume, such as the subsequent weighing of the column filled with two solvents of different densities [8], do not account for the preferential solvation of the stationary phase. Linearization of the logarithmic net retention times of the members of a homologous series for the determination of *V_m_* is a lengthy approach and presumes validity of the ideal reversed-phase model [11].

The assignment of the solvent adsorbed on the solid phase surface either to the stationary phase or to the mobile phase volume is still controversial [12]. Further, the exact position of the boundary (dividing plane) between the bulk mobile phase and the liquid occluded on the stationary phase is difficult, if possible at all [13]. The determination of the volume of the stationary phase is even more difficult in HILIC systems, because of the diffuse character of the stationary phase layer (water) in the inner pores of the column packing particles. According to the original Alpert model, the excess adsorbed water represents the stationary phase in HILIC [1]. Dinh et al. [14] presume that desorption and partitioning actually coexist as retention promoters for neutral solutes. Guo et al. [15] proposed determination of the sorbed water layer as the difference between the elution volumes of toluene in the aqueous/organic mobile phase and in pure acetonitrile. The distribution constant in LC depends on the retention mode and—deliberately or unintentionally—leads to the phase definition by accepting a convention.

***Convention 1 (classical)***: This is the standard convention used in the traditional determination of the retention factors in LC. The volume of the stationary phase, *V_s_*, is inaccessible to a non-retained marker compound neither retained in nor excluded from the stationary phase. This means that *V_s_* is equal to the empty column inner volume, *V_column_*, minus the total pore volume, *V_T_* (the inner pores, *V_i_*, plus the inter-particle pores, *V*_0_):*V_s_* = *V_column_* − *V_i_* − *V*_0_. In HILIC, a small non-polar hydrocarbon molecule (benzene or toluene) is the most frequent *V_m_* marker.

***Convention 2 (inner pore)***: In the (idealized) size-exclusion chromatography (SEC), the solid particle skeleton, *V_SKEL_*, is inert and does not participate in the distribution process. The full volume of the inner pores contains the liquid stationary phase, *V_s_* = *V_i_*, which has the same composition as the mobile phase in the inter-particle volume (such as tetrahydrofuran), *V_m_* = *V*_0_. The molecules of the solute distribute between *V_i_* and *V*_0_, based on their size, which controls the part of the accessible pore volume. The elution volume of a high-molecular weight standard for which the inner pores are inaccessible, (e.g., polystyrene with *M_r_* ≥ 10^6^), estimates the volume of the mobile phase, *V_m_*; the elution volume of a small hydrophobic molecule (benzene or toluene) estimates the volume of the sum of the inner and inter-particle pores, *V_i_* + *V*_0_.

***Convention 3 (adsorbed water)***: In HILIC—according to the original Alpert model—the sample distributes between the bulk, organic solvent-rich mobile phase and volume of water on the polar solid surface [1]. Hence, in the HILIC systems, the *convention 2*, setting the whole inner pore volume equal to the volume of the stationary phase in the column cannot be applied, as water (the liquid stationary phase) fills only a larger or a smaller part of the inner pore volume, in which it is diffused. The liquid in the inner pores differs from the bulk liquid mobile phase, contained in the inter-particle volume and in a part of the inner pores. The amount of the diffused water stationary-phase strongly depends on the composition of the aqueous–organic bulk mobile phase. The solid particle skeleton may contribute to the retention, mainly by the surface adsorption, and we can neglect its contribution to the retention at first approximation, so that we can understand the adsorbed diffuse layer of water as the stationary phase, *V_s_* = *V_ex_*, and the remaining liquid inside the pores and in the inter-particle volume as the mobile phase, *V_m_* = *V_i_* + *V*_0_ − *V_ex_*. Hence, the differences between the phase ratios defined by the two conventions depend on the composition of the mobile phase and are more significant for columns with stronger water uptake.

## 2. Experimental

### 2.1. Materials

Thiourea, uracil, toluene, phenol and the narrow distribution polystyrene standard (1,800,000 g/mol), were purchased from Merck (Merck KGaA, Darmstadt, Germany) in the best available purity. Acetonitrile, tetrahydrofuran (both LiChrosolvgradient grade), ammonium acetate and formic acid (both reagent grade) were obtained from Merck, Darmstadt, Germany. Water was purified using a Milli-Q water purification system (Millipore, Bedford, MA, USA). Coulomat^®^ AG, a reagent for coulometric Karl Fischer titration was purchased from Sigma Aldrich.

### 2.2. Equipment

Water adsorption was measured using an ECOM pump (ECOM, Prague, Czech Republic), connected to a chromatographic column and an automated fraction collector (CF-1 Fraction Collector, Spectrum Chromatography, Houston, TX, USA). Columns were placed in a thermostatted compartment set at 40 °C. A Karl Fischer titrator equipped with a Ti Stand magnetic stirrer (Metrohm, Herisau, Switzerland) was employed for the determination of the volume of water in the individual fractions The HPLC instrument was comprised of a high-pressure pump connected to a variable UV detector (both from ECOM). The columns were kept in a thermostatted column compartment at 40 °C.

### 2.3. HILIC Columns

A total of 18 polar columns for HILIC applications tested in the present work were selected to cover a broad range of bonded phase polarities. The selection includes two zwitterionic columns (sulfobetaine and phosphorylcholine), diol silica-based stationary phases (YMC Triart Diol, YMC, Kyoto, Japan; LiChrospher 100 DIOL, Merck, Darmstadt, Germany; Luna HILIC, Phenomenex, Torrance, CA, USA), ethylene-bridged hybrid silica (XBridge HILIC), silica (Atlantis HILIC and fused core-shell silica Ascentis Express HILIC), chemically bonded fused core-shell silica stationary phases (pentafluoro phenyl hexyl silica, silica with highly polar ligand possessing 5 hydroxyl groups, and silica modified with diisopropyl-cyanopropylsilane) and modified hydrosilated silica. Table 1 shows the basic manufacturer information on the dimensions and characteristics of the columns tested and the values of the column void volume conventionally measured as the elution volume of toluene.

### 2.4. Methods

The data necessary for the determination of the excess concentration of water, *q_ex_*, of the columns tested were acquired using the frontal analysis method. From the inflection points of the plots of water concentration in each fraction versus the volume of the mobile phase passed through the column (the breakthrough curves) the water breakthrough volumes, *V_B_*, were evaluated, followed by coulometric Karl Fischer titration: H_2_O + I_2_ + [RNH]^+^ SO_3_CH_3_^−^ + 2 RN →[RNH]^+^ SO_4_CH_3_^−^ + 2[RNH]^+^ I^−^(5)

First, each tested column was rinsed with 50 hold-up volumes ACN, the high-pressure pump and all capillaries were filled with mobile phase. The column was then disconnected, the pump and the connecting capillaries were filled with the feed solution of water in acetonitrile, the column was again connected to the system, and the fraction collector and the isocratic pump delivering the feed solution of water in acetonitrile onto the column were actuated. The column was kept in a thermostatted compartment at 40 °C during the frontal analysis and each feed solution was continuously delivered onto the column at a constant flow rate and fractions of the effluent passed through the column were collected at the rate of 5–12 fractions per minute. Twelve feed solutions of water in acetonitrile were passed through the tested column and the concentration of water in the collected fractions (0.08 mL each) was determined using the Karl Fischer method, and the concentration of water adsorbed on the stationary phase in excess of the concentration in the bulk mobile phase, *q_ex_*, was calculated for each concentration of water in the mobile phase, *c_m_*, as:(6)$$q_{ex}=\left(V_{B}-V_{m}\right)\frac{c_{m}}{V_{i}}$$

*V_m_* is the total volume of the liquid (i.e., the volumes volume of the inner pores, *V_i_* and of the interparticle volume, *V*_0_, in the column, respectively). An example of the method is illustrated for a ZIC-HILIC column in Figure 1. The approach was repeated in triplicate for each feed solution of water in acetonitrile, covering the range from 0.5% *v*/*v* to 15% *v*/*v* water on most columns tested and the range from 0.5% *v*/*v* to 40% *v*/*v* on the Ascentis Express CN column. The relative standard deviations of the experimental breakthrough volumes range between 0.32% and 0.76%, which also characterizes the error in the excess sorbed water amount.

## 3. Results and Discussion

For each column, the hold-up volume, *V_M_*, was measured using *convention 1* as the elution volume of toluene in 100% and 95% acetonitrile, respectively. The interstitial volume in between the particles, *V*_0_, was determined as the elution volume of the narrow-distribution high-molecular polystyrene standard with *M_r_* = 1,800,000 in 100% tetrahydrofuran. The total column porosity, *ε_T_*, was calculated from the volume of the empty column, *V_column_*, and the column hold-up volume, *V_M_*, (*ε_T_* = *V_M_*/*V_column_*), the interstitial column porosity as *ε_0_* = *V*_0_/*V_column_*. The inner porosity was obtained as the difference between the column total porosity and the interstitial porosity, *ε_i_* = *ε_T_* − *ε_0_*, i.e., the pore volume inside the column was calculated as the difference between the hold-up volume and the interstitial volume, *V_i_* = *V_M_* − *V*_0_. The dimensions and pore characteristics of 18 columns studied are listed in Table 1.

*V_s_* defined by *convention 3* strongly depend on the composition of the aqueous–organic bulk mobile phase. These effects are described by the isotherms of water on the individual columns. We combined a dynamic frontal analysis method with Karl Fischer titration as a direct method of determination of water in the collected effluent fractions [16]. The experimental isotherms of water adsorbed in the inner particle pores in excess of the bulk mobile phase concentration on the columns tested satisfactorily characterizes the Langmuir model, as shown in Equation (7) [17]:(7)$$q_{i}=\left(a\cdot c_{m}\right)/\left(1+\frac{a}{q_{s}}\cdot c_{m}\right)$$

*q_i_* is the volume fraction of water contained in the pores of the stationary phase in excess of the water contained in the bulk mobile phase, *c_m_* and *a* is the distribution constant of water in the pores of the stationary phase at a very low *c_m_*. *q_s_* in Equation (6) gives the maximum (saturation) column adsorption capacity for water. Figure 2 shows three examples of water isotherms on a zwitterionic ZIC-HILIC, a TSKgel amide and a hydrosilated Cogent Silica C columns [4].

The adsorption isotherms show the excess volume of water (beyond the water concentration in the bulk mobile phase) retained in a column. Table 2 shows large differences in the proportions of the inner pores occupied by the excess adsorbed water, calculated as *V_ex_* = *q_satur_*·*V*_0_, *V_ex_* (from 7 to 129 µL) for the 18 columns tested. The water sorption saturation capacities (in excess of the water concentration in the bulk mobile phase), *q_ex_* and the corresponding excess volume of water contained in the adsorbed diffuse layer, *V_ex_*. Even non-polar or slightly polar columns intended for reversed-phase LC applications (such as Cogent Cholesterol, Cogent Bidentate C18, Ascentis PhF5, Cogent Phenyl hydride) show a weak water adsorption. Due to a low affinity to water, hydrosilated silica stationary phases adsorb less than one monomolecular water layer equivalent (corresponding to 0.45 water monolayers for Cogent Silica C). Less than 9% *v*/*v* water in the mobile phase is sufficient to accomplish the full water saturation capacity of porous, core-shell, hybrid Ethylene Xbridge HILIC and hydrosilated silica gel columns, such as Cogent Silica C. On the other hand, the water adsorption isotherms on the ZIC HILIC and TSK gel amide 80 columns are shallow and even the mobile phases with 20% water/acetonitrile does not allow the achieving of full water saturation capacity (Figure 2). Columns with bonded polar ligands (hydroxyl, diol, nitrile, pentafluorophenylpropyl, diol, zwitterionic sulphobetaine and phosphorylcholine), and TSK gel amide and amine, show stronger water adsorption in comparison to the bare silica. At full column saturation, the amount of the excess adsorbed water, *V_ex_*, corresponds to 6–9 monomolecular equivalents of water layers, *N_w_*, for the ZIC cHILIC, and ZIC HILIC zwitterionic stationary phases, to 3–5 monomolecular water layer equivalents for the stationary phases with bonded hydroxyl groups (Luna HILIC, YMC Triart diol and Ascentis Express OH5) and to 1–2 water layer equivalents for the Xbridge HILIC and Atlantis HILIC columns [14]. 

The Table 2 and Figure 3 show large differences in the proportion of the volume of the column inner pores filled with the excess water (100*ε_H2O_*/*ε_0_*) in aqueous acetonitrile containing 15% water. In the fused-core shell silica columns, the nonporous core occupies a significant part of the void column volume; the excess adsorbed water, *V_ex_*, fills only 2.1% of the pore volume of the Ascentis Express Phenyl FS column, 8.5% of the Ascentis Express CN nitrile column, 1.7% of the Ascentis Express HILIC column and 3.5% of the Ascentis Express OH5 column. In the fully porous silica-based stationary phases containing –OH groups, LiChrospher 100 DIOL, YMC triart DIOL and Luna HILIC, the adsorbed excess water takes 2.4%, 15.1% and 11.3% pore volume, respectively. The hybrid organic-silica Xbridge HILIC and the Atlantis HILIC columns contain 5.3% and 9% adsorbed excess water, respectively, while the pore volume of less polar hydrosilated silica Cogent columns is filled by excess adsorbed water only to 2.7–5.4%. McCalley and Neue [7] published similar results suggesting that 4–13% of the pore volume of a silica-bonded phase is occupied by a water-rich layer in 95–75% (*v*/*v*) aqueous acetonitrile. Maximum water uptake in the acetonitrile with 15% water is observed for zwitterionic stationary phases ZIC HILIC (32%) and ZIC cHILIC (20%), and for TSK-gel Amide 80 (51%) and TSK-gel NH2 (46%).

Table 2 lists the porosities characterizing the proportions of the column volume corresponding to the inner pores, the inter-particle space and the volume occupied by the adsorbed water. The pore distribution is illustrated by the pie diagrams in Figure 4 for the ZIC HILIC, TSK Amide, Triart Diol, Luna HILIC and X-Bridge HILIC columns. Figure 5 compares the volume phase ratios in the 18 columns according to the two conventions. There are more or less significant differences in the column volume ratios of the stationary and the mobile phases, *V_s_*/*V_m_*, according to *convention 1* (stationary phase measured as the elution time of toluene) and *convention 3* (stationary phase as the amount of the excess uptake of water)—Table 2.

As the differences in the volumes of the mobile phase according to *convention 3*, *V_m_* = *V_i_* + *V*_0_ − *V_ex_*, differ (are larger by several per cent) from the *V_m_* calculated according to *convention 1*, *V_s_* = *V_ex_*, the differences in *k* are relatively more significant for the columns with larger proportions of the adsorbed water. The differences between the retention factors calculated by the two conventions depend on the composition of the mobile phase and are most significant for the zwitterionic and TSK gel columns. The differences are illustrated by several examples of the retention factors of polar phenolic compounds on the Lichrospher 100 Diol (*ε_H2O_* = 0.08) and Luna HILIC (*ε_H2O_* = 0.04) columns in 98% acetonitrile/water mobile phase in Table 3. As can be expected, the relative differences in the retention factors are larger (approximately + 12%) for the Diol column showing approximately twice higher adsorbed water amount in comparison to the Luna HILIC column (approximately + 5%). The examples in the table show that the adopted stationary phase convention does not affect the predicted separation selectivity of sample components (the relative retention, i.e., the ratio of the retention factors, *k_j_/k_i_*)—phenolic acids and flavones.

## 4. Conclusions

1. The determination of stationary and mobile phase volumes in HILIC (aqueous normal phase) separation systems can be simplified by adopting the convention that the excess water adsorbed by the column principally can be considered the stationary phase, in agreement with the original Alpert theory [1]. The volume of the aqueous stationary phase, *V_ex_*, can be determined experimentally by frontal analysis employing direct measuring the water concentration in small-volume fractions of the column effluent using the Karl Fischer titration method [16]. This method measures the excess water inside the column sorbed both by the partition and the adsorption processes in the case of a mixed retention mechanism.

2. The experimental isotherms of water adsorbed on polar columns allow direct predicting of the effects of the composition of aqueous–organic mobile phase on the retention in HILIC systems.

3. Columns with bonded polar ligands, especially zwitterionic and TSK gel amide and amine, show stronger water adsorption in comparison to bare silica. The column phase volume ratios calculated according to the new phase convention strongly depend on the type of the polar column. The volume of the liquid in the inner pores of the column, *V_i_*, includes both the stationary and the mobile phase.

4. For accurate determination of the retention factors, the volume of the mobile phase in the column, *V_m_*, should be corrected by subtracting the volume of the adsorbed water from the total volume of the liquid in the column, *V_m_* = *V_i_* + *V*_0_ − *V_ex_*. This correction provides larger retention factors than the commonly used determination of *V_m_* as the elution volume of small non-polar molecules, e.g., toluene. The differences between the corrected and uncorrected retention factors increase proportionally to the volume of the excess adsorbed water in the column. However, the correction does not affect the relative retention.

## Figures and Tables

**Figure 1 molecules-26-04819-f001:**
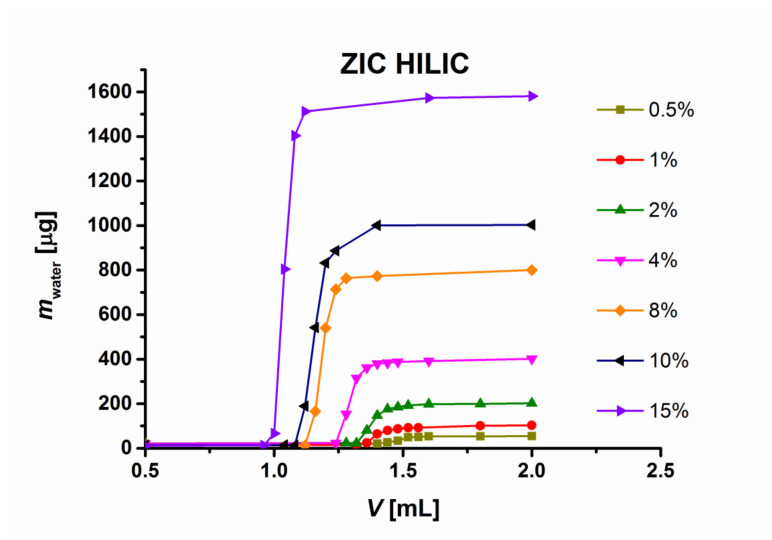
Frontal analysis (breakthrough) plots of water on ZIC HILIC stationary phase at 40 °C. *V*—volume of the mobile phase; *m_water_*—absolute weight of water in 10 µL fraction of mobile phase.

**Figure 2 molecules-26-04819-f002:**
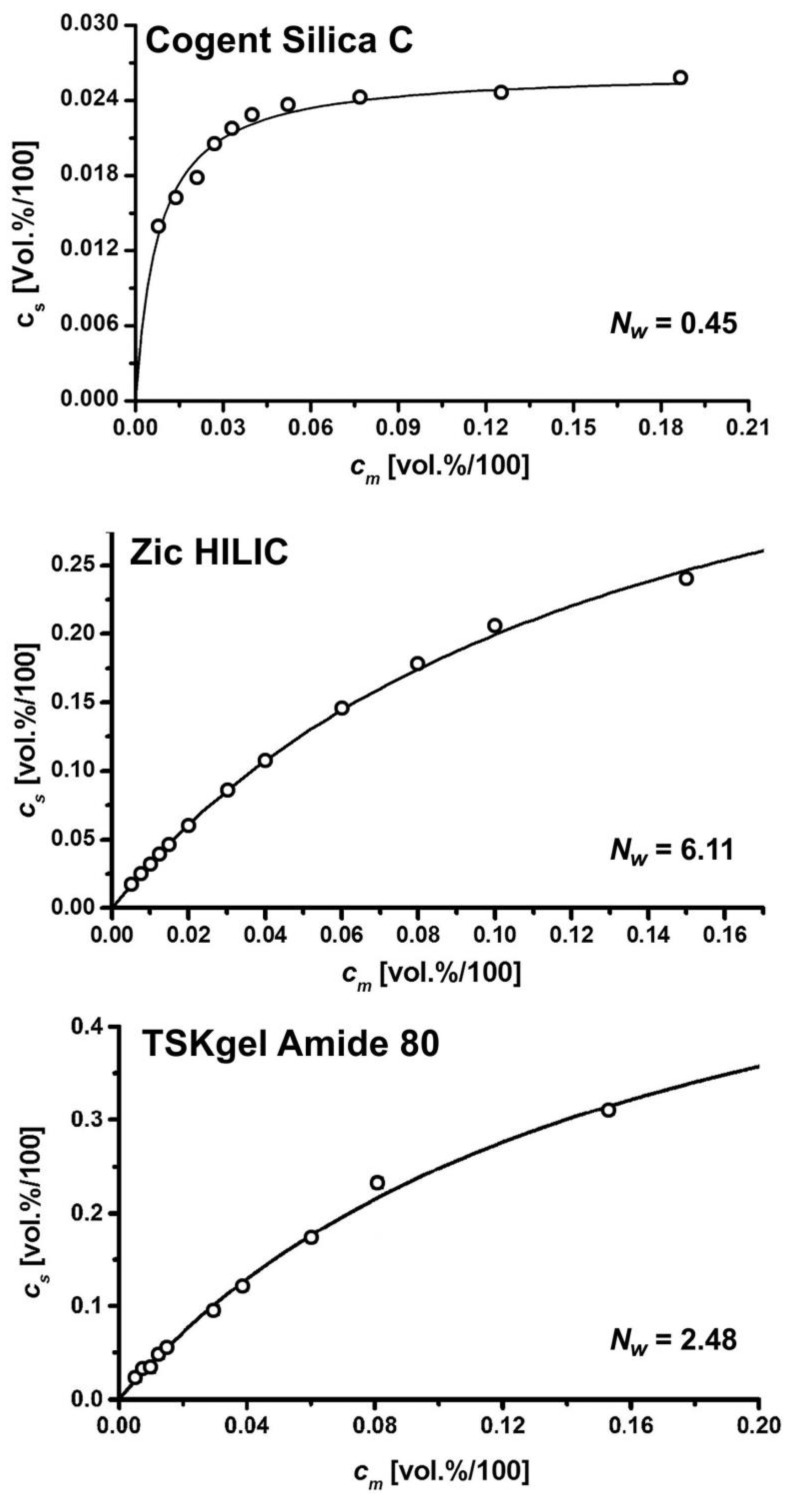
Sorption isotherms of water from acetonitrile on Cogent Silica C, ZIC HILIC and TSKgel Amide 80.

**Figure 3 molecules-26-04819-f003:**
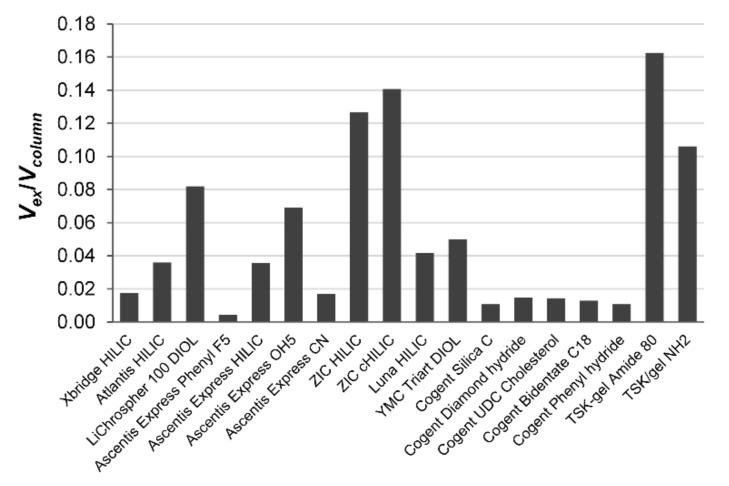
The volume ratios of the adsorbed layer of water (*V_ex_*) and volume of the 18 polar columns (*V_column_*).

**Figure 4 molecules-26-04819-f004:**
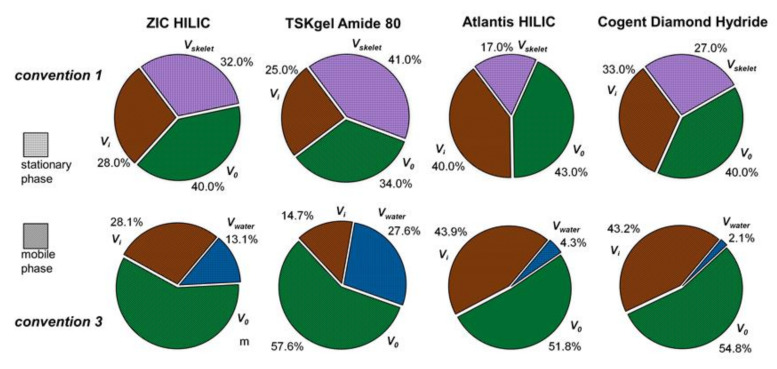
Distribution of the areas in the columns: skeleton, adsorbed water, inner pores, inter-particle space, stationary and mobile phases in four polar columns: Zic HILIC, TSKgel Amide 80, Atlantis HILIC and Cogent Diamond Hydride.

**Figure 5 molecules-26-04819-f005:**
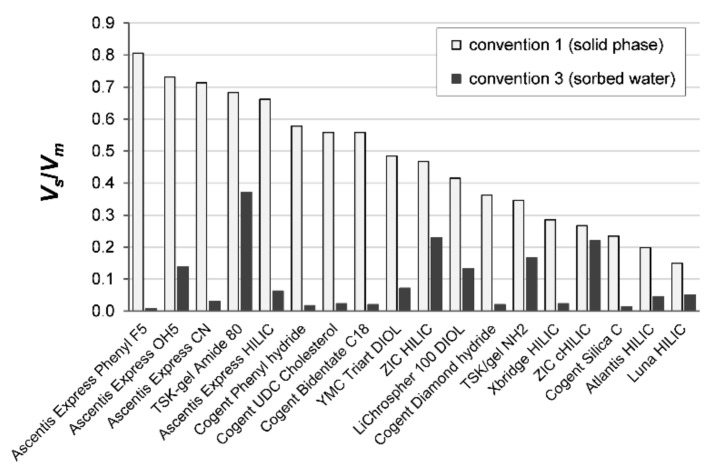
The volume ratios of the stationary (*V_s_*) and mobile (*V_m_*) phases according to convention 1 (stationary phase = solid phase)—blank blocks—and convention 3 (stationary phase = sorbed water diffuse layer)—black blocks—for 18 polar columns.

**Table 1 molecules-26-04819-t001:** Types and dimensions of the columns tested.

Commercial Name	Type of Stationary Phase	*L* (mm)	*i*.*d*. (mm)	*V_column_* (mL)	Particle Size (μm)	*V*_M_ (mL)	*ε* _T_	*ε* _0_	*ε* _i_	*V*_i_ (mL)
Xbridge HILIC	Ethylene bridge silica	100	3.0	0.71	5.0	0.55	0.77	0.43	0.34	0.19
Atlantis HILIC	Bare silica	100	3.0	0.71	5.0	0.59	0.83	0.43	0.40	0.24
LiChrospher 100 DIOL	Diol silica	125	4.0	1.67	5.0	1.11	0.70	0.37	0.33	0.37
Ascentis Express Phenyl F5	Fused core pentafluorphenylpropyl silica	100	4.6	1.66	2.7	0.92	0.55	0.36	0.19	0.17
Ascentis Express HILIC	Fused core shell silica	100	4.6	1.66	2.7	1.00	0.60	0.39	0.21	0.21
Ascentis Express OH5	Fused core pentahydroxyl silica	100	4.6	1.66	2.7	0.96	0.57	0.37	0.20	0.33
Ascentis Express CN	Fused core diisopropyl-cyanopropylsilica	100	4.6	1.23	2.7	0.97	0.58	0.38	0.20	0.19
ZIC HILIC	Zwitterionic sulfobetaine	250	2.1	0.86	3.5	0.59	0.68	0.40	0.28	0.16
ZIC cHILIC	Zwitterionic phosphorylcholine	150	2.1	0.52	3.0	0.41	0.78	0.48	0.30	0.12
Luna HILIC	Crosslinked diol silica	150	2.0	0.47	3.0	0.41	0.88	0.49	0.39	0.16
YMC Triart DIOL	Organic/silica dihydroxypropyl hybrid	150	2.1	1.25	5.0	0.35	0.75	0.42	0.33	0.12
Cogent Silica C	Hydrosilated silica	75	4.6	1.01	4.0	1.01	0.81	0.40	0.41	0.41
Cogent Diamond hydride	Hydrosilated carbon silica	100	4.6	1.25	4.0	1.22	0.73	0.40	0.33	0.40
Cogent UDC Cholesterol	Hydrosilated cholesteryl silica	75	4.6	1.25	4.0	0.80	0.64	0.38	0.26	0.21
Cogent Bidentate C18	Hydrosilated octadecyl silica	75	4.6	1.25	4.0	0.80	0.64	0.39	0.25	0.16
Cogent Phenyl hydride	Hydrosilated phenyl silica	150	4.6	2.49	4.0	1.58	0.63	0.36	0.28	0.44
TSKgel amide 80	Bonded carbamoyl groups on silica	150	2.0	0.47	3.0	0.28	0.60	0.34	0.25	0.18
TSKgel NH_2_	Bonded aminopropyl groups on silica	150	2.0	0.47	3.0	0.35	0.74	0.40	0.40	0.33

*L*—column length (mm); *i*.*d*.—column inner diameter (mm); *V_column_*—geometric volume of the empty column (mL); *V*_M_—volume of the mobile phase in the column measured as the elution volume of toluene (mL); *ε*_T_—total porosity of the inner pores *V_i_* and of the inter-particle volume *V*_0_, from the elution volume of toluene in acetonitrile; *ε*_0_—inter-particle column porosity measured from the elution volume of a polystyrene standard (1,800,000 g/mol); *ε_i_*—inner porosity determined as the difference between the total and the inter-particle porosity, *ε_i_ =*
*ε*_T_ − *ε*_0_.

**Table 2 molecules-26-04819-t002:** Porosities and volumes of the stationary and mobile phases in 18 polar HPLC columns according to different conventions.

Commercial Name	*V_column_* (mL)	*V*_M_ (mL)	*V*_i_ (mL)	*V*_0_ (mL)	*ε* _T_	*ε* _i_	*ε* _0_	*q_ex_* (*v*/*v*)	*ε_H20_*	V_ex_ (µL)	*100ε_H2_*_0/_*ε*_i_ (%)	*V_s_/V_m_* Conv. 1	*V_s_/V_m_* Conv. 3
Xbridge HILIC	0.71	0.55	0.19	0.36	0.77	0.34	0.43	0.054	0.018	12.8	5.3	0.285	0.023
Atlantis HILIC	0.71	0.59	0.24	0.35	0.83	0.40	0.43	0.089	0.036	25.4	9.0	0.198	0.045
LiChrospher 100 DIOL	1.57	1.11	0.37	0.74	0.70	0.33	0.37	0.248	0.081	128.7	2.4	0.415	0.131
Ascentis Express Phenyl F5	1.66	0.92	0.17	0.75	0.55	0.19	0.36	0.023	0.004	7.1	2.1	0.806	0.004
Ascentis Express HILIC	1.66	1.00	0.21	0.79	0.60	0.21	0.39	0.168	0.035	58.9	1.7	0.662	0.062
Ascentis Express OH5	1.66	0.96	0.33	0.63	0.57	0.20	0.37	0.354	0.069	115.1	3.5	0.731	0.136
Ascentis Express CN	1.66	0.97	0.19	0.78	0.58	0.20	0.38	0.083	0.017	28.2	8.5	0.713	0.030
ZIC HILIC	0.86	0.59	0.16	0.43	0.68	0.28	0.40	0.453	0.089	109.6	31.8	1.085	0.228
ZIC cHILIC	0.52	0.41	0.12	0.29	0.78	0.30	0.48	0.456	0.060	73.1	20.0	0.146	0.217
Luna HILIC	0.47	0.41	0.16	0.25	0.88	0.39	0.49	0.105	0.041	19.6	11.3	0.149	0.050
YMC Triart DIOL	0.52	0.35	0.12	0.23	0.75	0.33	0.42	0.169	0.050	26.0	15.1	0.484	0.026
Cogent Silica C	1.25	1.01	0.41	0.60	0.81	0.41	0.40	0.026	0.011	13.4	2.7	0.234	0.016
Cogent Diamond hydride	1.25	1.22	0.40	0.50	0.73	0.33	0.40	0.044	0.015	24.4	4.5	0.362	0.020
Cogent UDC Cholesterol	1.25	0.80	0.21	0.48	0.64	0.26	0.38	0.055	0.014	18.0	5.4	0.558	0.028
Cogent Bidentate C18	1.25	0.80	0.16	0.50	0.64	0.25	0.39	0.051	0.013	16.0	5.2	0.558	0.035
Cogent Phenyl hydride	2.49	1.58	0.44	0.90	0.63	0.28	0.36	0.039	0.011	26.8	3.9	0.578	0.011
TSK-gel Amide 80	0.47	0.28	0.12	0.16	0.60	0.25	0.34	0.265	0.163	63.9	51.2	0.679	0.337
TSK/gel NH_2_	0.47	0.35	0.04	0.31	0.74	0.34	0.40	0.319	0.157	73.8	46.1	0.343	0.166

*V_column_*—geometric volume of the empty column (mL); *V*_M_—volume of the mobile phase in the column measured as the elution volume of toluene (mL); *ε*_T_—total porosity of the inner pores *Vi* and of the inter-particle volume *V*_0_, from the elution volume of toluene in acetonitrile; *ε*_0_—inter-particle column porosity measured from the elution volume of a polystyrene standard (1,800,000 g/mol); *ε_i_*—inner porosity determined as the difference between the total and the inter-particle porosity, *ε_i_ =*
*ε*_T_ − *ε*_0_. *V_column_*, *q_ex_* (*v*/*v*)—column excess saturation capacity (Equation (7)); *ε_H20_*—the part of the column porosity occupied by the sorbed water; V_ex_ (µL)—volume of the sorbed water in the column in water/acetonitrile 15: 85 (*100ε_H2_*_0/_*ε*_i_ %), part of the inner pore volume occupied by sorbed water; *V_s_/V_m_* (Conv. 1)—volume ratio of the stationary to the mobile phase according to convention 1 (*V_m_—elution volume of toluene*); *V_s_/V_m_* (Conv. 3)—volume ratio of the stationary to the mobile phase according to convention 3 (*V_s_—volume of the excess adsorbed water*).

**Table 3 molecules-26-04819-t003:** Effects of the stationary phase volume *convention 1* ((1)—solid phase)) and *convention 3* ((3)—sorbed water)) on the hold-up volume, *V_m_*, retention factors, *k_i_* and selectivity (relative retention, *k_j_/k_i_*) on the Lichrospher100 Diol and Luna HILIC columns in 98% acetonitrile.

	Lichrospher 100 Diol		Luna HILIC	
**Mobile Phase Volume**	*V_m_* (1) = 1.11	*V_m_* (3) = 0.98	*V_m_* (1) = 0.41	*V_m_* (3) = 0.39
**Compound**	*k* (1)	*k* (3)	*k* (1)	*k* (3)
Gallic acid	4.42	4.99	2.19	2.30
Protokatechuic acid	1.72	1.94	1.02	1.07
Salicylic acid	1.28	1.45	1.32	1.39
Caffeic acid	1.16	1.31	0.91	0.96
Coumaric acid	0.71	0.80	0.40	0.42
Catechine	1.24	1.40	1.11	1.17
Rutine	15.16	17.13	8.53	8.97
Naringine	5.33	6.02	3.00	3.15
Hesperidine	4.12	4.66	2.18	2.29
*k*_Naringine_/*k*_Hesperidine_	1.29	1.29	1.38	1.38
*k*_Gallic_/*k*_Protokatechuic_	2.57	2.57	2.15	2.15
*k*_Salicylic_/*k*_Caffeic_	1.10	1.11	1.45	1.45

## Data Availability

The data presented in this study are available on request from the authors.

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
