# Peer review of "A New Definition of the Stationary Phase Volume in Mixed-Mode Chromatographic Columns in Hydrophilic Liquid Chromatography"

_molecules, 2021, doi:10.3390/molecules26164819_

Round 1

Reviewer 1 Report

To Authors:

  • The manuscript named “A new definition of the stationary phase volume in mixed mode chromatographic columns in Hydrophilic Liquid Chromatography” is an interesting piece of work.
  • This work addresses gaps in analytical methods available to quantify adhered water layer in the proposed HILIC mechanism. This topic is a known challenging topic and many scientists have brought up several approaches to obtain accurate quantification of water layer adhered to the SP. In this manuscript, the authors provide a much depth analysis of the structure.
  • The abstract is precise and meaningful.
  • The introduction provides a good background understanding.
  • The rest of the content is well presented. The references are carefully chosen, and they are very appropriate to the topic. Please see my comments below regarding references.
  • The presentation of data/information is clear and readable.
  • Figures are well arranged. Tables are clear and informative. Please see my comments below regarding the Figures and Tables.
  • Discussion: Observations are well described with the support of scientific evidence.
  • Conclusion: Well supported by results.

Overall, this manuscript is recommended to publish after few minor revisions listed below.

  1. Line 9: Please define HILIC in Abstract.
  2. Line 55: Please define LC.
  3. Line 97,98: It is suggested to reword the sentence “The ratio of the column length, L, controls the linear velocity of the mobile phase along the column, u” for better readability.
  4. Line 131: the The distribution constant……
  5. Line 163: Please define Vex in the introduction section as well.
  6. What is the reason to choose 40°C?
  7. It is suggested to include any specific Karl fisher conditions in the experimental sections.
  8. Table 1: Please define abbreviations used in the column headings along with the abbreviations for better readability of the Table.
  9. In Table 1, my understanding is Vcolumn represents the theoretical column volume – π x r2 x L. Based on this scenario, some Vcolumn numbers do not match with the theoretical number?

LiChrospher 100 DIOL

Ascentis Express CN

YMC Triart DIOL

Cogent Silica C

Cogent UDC Cholesterol

  1. Please replace Figure 2 with a high-resolution image for better readability.
  2. It is suggested to describe error associated with water determination by KF and how it affects the results.
  3. Below 2 references are suggested to add appropriately into the introduction or discussion section.                                                    Dinh, Ngoc Phuoc, Tobias Jonsson, and Knut Irgum. "Water uptake on polar stationary phases under conditions for hydrophilic interaction chromatography and its relation to solute retention." Journal of Chromatography A 1320 (2013): 33-47.                                              Guo, Yong, et al. "Evaluating the adsorbed water layer on polar stationary phases for hydrophilic interaction chromatography (HILIC)." Separations 6.2 (2019): 19.

Author Response

  1. Line 9: Please define HILIC in Abstrakt  Hydrophilic Interaction Liquid Chromatography. Done.

  1. Line 55: Please define LC  Liquid Chromatography. Done.
  2. Line 97,98: It is suggested to reword the sentence “The ratio of the column length, L, controls the linear velocity of the mobile phase along the column, u” for better readability. Changed to: The ratio of the column length, L, to the column hold/up time, tm, controls the linear velocity of the mobile phase along the column, u.
  3. Line 131: the The distribution konstant   OK……
  4. Line 163: Please define Vex in the introduction section as well.   . According to the original Alpert model, the excess adsorbed water represents the stationary phase in HILIC.
  5. What is the reason to choose 40°C?  This temperature is better controlled than 20 – 25o C by older column thermostats and is low enough not to cause a degradation of the stationary phase.
  6. It is suggested to include any specific Karl fisher conditions in the experimental sections. The conditions are described in the experimental section 2.4.
  7. Table 1: Please define abbreviations used in the column headings along with the abbreviations for better readability of the Table. Done.
  8. In Table 1, my understanding is Vcolumn represents the theoretical column volume – π x r2 x L. Based on this scenario, some Vcolumn numbers do not match with the theoretical number?

LiChrospher 100 DIOL

Ascentis Express CN

YMC Triart DIOL

Cogent Silica C

Cogent UDC Cholesterol

The data was checked and corrected where necessary.

  1. Please replace Figure 2 with a high-resolution image for better readability. Done.
  2. It is suggested to describe error associated with water determination by KF and how it affects the results.  The relative standard deviations of the experimental breakthrough volumes range in between 0.32% to 0.76%, which characterizes also the error in the excess sorbed water amount..
  3. Below 2 references are suggested to add appropriately into the introduction or discussion section.                                                    Dinh, Ngoc Phuoc, Tobias Jonsson, and Knut Irgum. "Water uptake on polar stationary phases under conditions for hydrophilic interaction chromatography and its relation to solute retention." Journal of Chromatography A 1320 (2013): 33-47.                                              Guo, Yong, et al. "Evaluating the adsorbed water layer on polar stationary phases for hydrophilic interaction chromatography (HILIC)." Separations 6.2 (2019): 19. The references were added as suggested – new refs. 14, 15..

Reviewer 2 Report

The is an excellent methods article, and it is exceedingly well written.  It was a pleasure to read it.  It should be noted, however, that Figure 2 impossible to read, and it must be revised.  It did not appear at all in the copy of the MS that I downloaded, and I downloaded it twice.

Author Response

Fig.2 has been uploaded again.

Reviewer 3 Report

This article dressed the water absorption in HILIC column altered the volume of stationary phase of a column and will influence the retention of analytes.

Some column require specific pH for better performance, did the author test all 17 columns in the same pH or not buffered?

Fig 2 is unclear, please consider to replace.

The excess volume of water over the concentration of mobile phase is due to the absorbance of water in the stationary phase (bounded water) and was obtained by calculation. Therefore, in Table 2, the sum of Vm+Vi+V0 will over V column. This is truth, would you explain the absorbed water was contributed to Vi or/and Vm and their portion?.

The phenomena of water absorbed in stationary phase and contribute to retention is very important. Would the author put more words in the application of the phenomena in chromatography and it importance?

Author Response

Fig. 2 has been uploaded again.

The meaning of the symbols has been explained more in detail in Tables 1 and 2. In Table 2, the sum of Vm = Vi+V0 . This was checked,

The absorbed water was contributed to Vi only (inner pores).

I agree that the phenomena of water absorbed in stationary phase and contribute to retention is very important, not only in HILIC. However a detailed discussion of this topic would go byond the size of the present work.